# Rapid Sequestration of Ecosystem Carbon in 30-year Reforestation with Mixed Species in Dry Hot Valley of the Jinsha River

**DOI:** 10.3390/ijerph16111937

**Published:** 2019-05-31

**Authors:** Zhilian Gong, Ya Tang, Wenlai Xu, Zishen Mou

**Affiliations:** 1Department of Environment Engineering, College of Food and Biological Engineering, Xihua University, Chengdu 610039, China; 0120020092@mail.xhu.edu.cn; 2Department of Environment, College of Architecture and Environment, Sichuan University, Chengdu 610065, China; nstxdy@163.com; 3State Key Laboratory of Geohazard Prevention and Geoenvironment Protection, Chengdu University of Technology, Chengdu 610059, China; 4Haitian Water Grp Co Ltd., Chengdu 610059, China

**Keywords:** deep soil organic carbon, total ecosystem carbon sequestration, mixed plantation, biomass carbon

## Abstract

Reforestation plays an important role in the carbon cycle and climate change. However, knowledge of ecosystem carbon sequestration through reforestation with mixed species is limited. Especially in dry hot valley of the Jinsha River, no studies cover total ecosystem carbon sequestration level in mature mixed plantations for a limited area of mixed plantations and difficulty in the sampling of plant roots and deep soil. In this study, carbon sequestration of seven mixed plantations of different ages in dry hot valley of the Jinsha River was investigated with analogous sites method. The results are as follows: 1) Deep soil organic carbon (SOC) storage significantly increased with stand age (*p* = 0.025), possibly due to fine root exudates and dissolved organic carbon transportation into deep soil and retention. 2) Total biomass carbon storage in the 30-year-old mixed plantation was 77.78 t C ha^−1^, 54 times reference wasteland and 9 times reference natural recovery shrub-grassland. However, total biomass carbon storage of 30-year-old mixed plantation was insignificantly lower than that of reference natural forest (*p* = 0.429). After 30 years of reforestation, plantation biomass carbon storage recovered to reference level, and its sequestration rate was 2.54 t C ha^−1^ yr^−1^. 3) The total ecosystem carbon storage of 30-year-old mixed plantation was 185.50 t C ha^−1^, 2.38 times reference wasteland, 2.29 times reference natural recovery shrub grassland, and 29% lower than reference natural forest. It indicated that niche complementary, good stand structure, and characteristics of dominant species *Leucaena leucocephala* in mixed plantations facilitate more rapid carbon sequestration, especially biomass carbon in the dry hot valley.

## 1. Introduction

Reforestation is extremely important to influence global climate for changing carbon sequestration of soil and biomass [1]. Because soil organic carbon (SOC) sequestration is influenced by various factors, such as climate, soil type, forest type, and stand age, large variations have been reported in the direction and amount of SOC stock change following reforestation [2,3,4]. Due to difficulty in collecting deep soil samples, especially from forestland [5], the organic carbon in deep soil layers has been poorly studied. However, deep soil (at 20–100 cm depth) stores 50–67% of SOC in one-meter depth, an important proportion of the soil C pool [6,7]. Deep SOC (below 20 cm) accumulation may be more important for enhancing carbon sequestration than topsoil SOC accumulation [8,9]. Shi et al. (2013) also found that the average SOC accumulation at soil depths of 0–60 cm and 0–100 cm was 49% and 55%, respectively, greater than the SOC accumulation at 0–20 cm depth of corresponding fields [10]. The SOC change at depths below the top 20 cm of reforested croplands contributed 28.4–46.6% of the total SOC change in one meter. Because of intimate association with silt and clay particles and deep soil (below 20 cm) environment that is favorable to soil carbon long-term storage, large quantities of organic carbon was stored in deep soil [8,11]. Therefore, to evaluate the effects of reforestation on SOC, it is necessary to investigate variation in organic carbon in deep soil layers.

Biomass plays an important role in carbon sequestration during reforestation. Lewis et al. (2019) indicated that the greatest potential for C sequestration associated with reforestation was through above- and below-ground tree biomass sequestration in the subtropical area after reforestation on croplands [12]. Tree biomass carbon sequestration rate was 6.1 t C ha^−1^ yr^−1^. Zhang et al. (2012) reported that biomass carbon stocks increased with stand age, reaching, respectively, 104 t C ha^−1^ and 94 t C ha^−1^ in mature *Eucalyptus* and *Acacia* plantations in south China [13]. In the subtropical area, biomass carbon sequestration rate of 19-year-old plantation on former wasteland can be up to 1.3–2.3 t C ha^−1^ yr^−1^ [14].

It is necessary to study total ecosystem carbon to know about the overall impact of reforestation on carbon pool structure and the carbon cycle. However, knowledge of the total ecosystem carbon accumulation during reforestation is limited. Wei et al. (2013) reported that the total ecosystem carbon stock in the restoration sites was 60% higher than that in the experimental controls after 19 years of reforestation on degraded wastelands, and they also demonstrated that active restoration could enhance ecosystem carbon sequestration in subtropical area. It also indicated that there were significant differences among component carbon pools within restoration functional groups [14]. Xie et al. (2013) found that ecosystem carbon pool in 24-year-old *Pinus massoniana* plantation was 10 times higher than that in degraded bare land and 22% lower than that in secondary forest, and they indicated that reforestation could facilitate rapid accumulation of carbon on severely eroded red soils through reforestation in subtropical China [15]. Lewis et al. (2019) suggested that it would take nearly 50 years for C stocks of Australian subtropical plantation to reach a similar level of natural ecosystems, with carbon sequestration rate of 7.4 t C ha^−1^ yr^−1^ [12]. 

Studies in dry hot valley of the Jinsha River have focused exclusively on either carbon in biomass or soil in isolation in young plantations. No studies cover total ecosystem carbon sequestration level in mature mixed plantations, including SOC, tree biomass carbon, and secondary biomass carbon (biomass carbon of shrubs, grasses, litter, and coarse woody debris defined as dead trees and broken branches with a base diameter more than 1 cm) and its relative contribution compared with remnant natural forests for limited area of mixed plantations and natural forest and difficulty in sampling plant roots and deep soil. People have tried to reforest on wastelands to improve the ecosystem structure and function and finally replicate natural forest, but mixed plantations with trees and shrubs were not common because pure plantations were usually implemented for reforestation in the dry hot valley. Besides, carbon stocks of remnant natural forest are not quantified because natural forests below 1600 m above sea level hardly exist for bad anthropogenic disturbance except few protected ones, so-called “Fengshui” forests. Studying C storage of remnant natural forest is important to know historical C losses through deforestation and sequestration potential following reforestation. Whether total ecosystem C stocks can recover to a reference level by reforestation hasn’t been studied in the dry hot valley. Natural recovery has also not been common in the dry hot valley because most people think it restores slowly.

In the present study, seven mixed plantations of different stand ages (between 9 and 30 years) in dry hot valley of the Jinsha River were selected, with adjacent wastelands, a natural forest, and a natural recovery shrub grassland as comparisons. SOC in 0–80 cm soil profile or deep into rock bed was measured for all stands. Biomass carbon (trees, shrubs, grasses, litter, and coarse woody debris) was measured for 30-year-old mixed plantation, adjacent reference wasteland in Hulukou town, the reference natural forest, and the reference natural recovery shrub-grassland. The objectives of this study were: 1) to evaluate the impact of 30 years of reforestation on former wastelands with mixed species on SOC sequestration in 0–80 cm soil profile in dry hot valley of the Jinsha River and the contribution of deep SOC (20–80 cm depth) sequestration; 2) to evaluate total ecosystem carbon sequestration of 30-year-old mixed plantation compared with remnant natural forests to get the latent capacity of ecosystem carbon sequestration of mature mixed plantation in dry hot valley and to know carbon sequestration contribution of soil and biomass. We hypothesized that 1) surface and deep SOC storage increased with stand age and deep SOC sequestration contribution to main part of total SOC sequestration; 2) after 30 years of reforestation, total ecosystem carbon, including SOC and biomass carbon, had recover to reference level of natural forest, and SOC sequestration contribution accounted for most. 

## 2. Materials and Methods

### 2.1. Study Area and Sites

This study was conducted in parts of Ningnan county of Sichuan province and Dongchuan municipality of Yunnan province, which is located in dry hot valley of the lower Jinsha River in southwestern China. The climate, soil type, and vegetation have been reported in the former studies in this area [16,17]. Seven mixed plantations of different ages (9–30 years) established with *Leucaena leucocephala* and other species, were selected, six in Ningnan and one in Dongchuan (Table 1). The situations of the mixed plantations, reference wastelands, reference natural recovery shrub grassland, and reference natural forest have been described in the former study of species diversity [17]. We used analogous sites (spatial) in place of temporal chronosequence with a limitation of ensuring forest stands of different ages along the identified chronosequence having similar soils and land use histories. But all sampling stands were in the dry hot valley, and we took paired measurements of the adjacent reference wastelands to account for differences in soil type and land-use history among properties. Details of the study sites are presented in Table 1. Lands of mixed plantations and reference natural recovery shrub grassland had not been cultivated for at least 20 years prior to reforestation and natural recovery, respectively. All reference wastelands were 400–600 m away from each corresponding mixed plantation. For limited area of natural forests and natural recovery shrub grasslands with similar topography and edaphic conditions to those of mixed plantations in dry hot valley, only one reference natural recovery shrub grassland (located in Hulukou town, and about 1.0–1.5 km away from adjacent mixed plantations) and one reference natural forest (located in Pisha town, and about 2.5 km away from adjacent mixed plantations) were selected. Although to some extent there is a limitation for lack of a reference natural recovery grassland and a reference natural forest for each mixed plantation, they were located in the same area (dry hot valley) as all the mixed plantations in this study. The soil is dry red soil (Chromic Cambisol in Food and Agriculture Organization (FAO) taxonomy, Ustochrept in United States Department of Agriculture (USDA) taxonomy). Mean annual temperature is 20–22 ℃, and average annual rainfall is 700–800 mm.

Three 10 m × 10 m plots were selected randomly in each mixed plantation, each reference wasteland, and reference natural recovery shrub grassland for data collection. Three 20 m × 20 m plots were selected randomly from the reference natural forest stand. Each plot of reference stands has similar topography and soil conditions to corresponding adjacent mixed plantation plots (Table 1 and Table 2). For limited stand area, plots at the 9-year-old mixed plantation and corresponding adjacent reference wasteland were about 100 m away from each other within a stand. The distance among plots within a stand at the other sampling stands was 200–300 m. 

### 2.2. Sampling and Analysis

Because the soil was about 90 cm deep in all stands except reference natural recovery shrub grassland (with a soil depth of about 60 cm), three soil pits were excavated to 80 cm depth or to bedrock (in reference natural recovery shrub grassland) randomly in each plot. After removing the litter, soil samples were taken by a shovel from depths of 0–20, 20–40, 40–60, and 60–80 cm, respectively, along the soil profile. At every soil layer, 500 g soil sample from each of the three soil pits were bulked to one composite sample for each soil depth in each plot. The adjacent reference wastelands, reference natural recovery shrub grassland, and reference natural forest were sampled as the reference sites (Table 1) to emulate soil properties prior to reforestation, under natural succession conditions and at an optimum state of mixed plantations, respectively. In the laboratory, soil samples were air dried, and pebbles and plant roots larger than 2 mm were filtered out with a screen (2 mm mesh). Soil bulk density was measured using a stainless steel bucket at the same depth range [18]. Soil samples for bulk density were sampled from the wall of the pits by a 100 cm^3^ bucket for each soil layer, put into labeled airtight plastic bags, and sent into the laboratory to be dried. Bulk density was calculated by dividing the oven-dry mass by the volume of the bucket. SOC concentration was analyzed with the modified Walkley-Black method [18]. The soil pH was measured with a combination electrode (soil-to-water ratio of 1:2) [18]. SOC storage for each soil layer was calculated by multiplying mean C concentrations by bulk density and soil depth, and total SOC storage in the soil profile was finally estimated by summing SOC storage of all soil layers as described by Wei et al. (2011) [19]. SOC accumulation rate in each mixed plantation was calculated as the difference in SOC storage between each mixed plantation and the reference wasteland, divided by the recovery time.

Biomass carbon was investigated in the stands of 30-year-old mixed plantation, adjacent reference wasteland, reference natural recovery shrub grassland, and reference natural forest. Plots were set as adjacent soil sampling plots of the same plot size. The aboveground and root biomass of the tree layer was estimated by the mean tree technique [20]. In order to determine the height and diameter at breast height of the mean tree, all trees in the plots were measured for height and diameter at breast height (DBH). Six sample trees that best represent the mean size collected at random from all the plots in the mixed plantation and reference natural forest, respectively, were destructively sampled, including stems, branches, foliage, and roots. The mean tree was cut at the ground level. Stems were cut into 1 m sections until the crown. At the end of each stem section, a 5 cm thick disc was sampled. The crown was divided into branches and foliage. Stems, branches, and foliage were weighed to get the fresh weight. In order to assess root biomass, a 1.0 × 1.0 m^2^ centered around each sample tree in the 30-year-old mixed plantation and a 1.5 × 1.5 m^2^ in the reference natural forest was excavated to a depth of 80 cm. The roots were collected, washed, air-dried, and divided into the stump, coarse (> 5 mm), and fine (< 5 mm) roots. The shrub biomass (including above-ground biomass and roots) was estimated by destructive sampling of three 2 × 2 m subplots, and grass biomass within three 1 m^2^ subplots along the diagonal of each plot. Roots of shrub and grass were collected to a soil depth of 30 cm in the subplots. The coarse woody debris (defined as dead trees and broken branches with a base diameter more than 1 cm; broken branches with a base of diameter no more than 1 cm were collected as litter) was collected within two 5 × 5 m subplots, and litter within three 1 m^2^ subplots along the diagonal of each plot. From each component, approximately 500–1000 g of fresh mass were randomly sampled and placed in a labeled airtight bag for moisture content determination if the fresh weight of the component mass was greater than 1000 g; otherwise, the whole component biomass was sampled for moisture content determination. All components and samples obtained were field weighed, and samples were kept cool until they could be transported to the laboratory. In the laboratory, the samples were dried to a constant mass at 85 ℃ in forced-air ovens. The carbon content was estimated as 50% of the biomass values [15]. 

### 2.3. Statistical Analyses

Data were analyzed after testing for normality and homogeneity of variance. One-way ANOVAs followed by least significant difference (LSD) tests was used to analyze the differences of SOC storage, biomass carbon storage, and total carbon storage among different vegetation types. Pearson correlation analysis was used to test the relationships between SOC storage and stand age. Analyses were performed using SPSS 20.0 for Windows. Significance levels were set at α = 0.05 in all statistical analyses. 

## 3. Results

### 3.1. Changes in Soil Organic Carbon Sequestration with Depth 

Surface soil (0–20 cm) organic carbon storage of mixed plantations ranged from 32.55 to 64.89 t C ha^−1^, with an average value of 46.31 t C ha^−1^ (Figure 1). Compared to the reference wasteland, SOC storage increased significantly at 0–20 cm depth after 9 years of reforestation, at 20–40 cm depth after 14 years of reforestation, and at 40–60, 60–80 cm depth after 26 years of reforestation. Compared to reference natural recovery shrub grassland, SOC storage of mixed plantations was significantly higher at 0–20 cm depth at the age of 14 years old and at 20–40, 40–60 cm depth at the age of 26 years old. SOC storage of mixed plantations was significantly lower than that of reference natural forest for all soil layers. 

Total SOC storage in 0–80 cm soil profile of mixed plantations (9–30 years old) ranged from 79.27 to 137.61 t C ha^−1^, 1.09–1.70 times the reference wasteland and 43–75% of reference natural forest (183.34 t C ha^−1^). SOC storage (0–60 cm) of mixed plantations was significantly higher than that of reference natural recovery shrub grassland after 26 years of reforestation (*p*= 0.000). Deep SOC storage in 20–80 cm depth of mixed plantations ranged from 46.71 to 72.72 t C ha^−1^, 53–60% of total SOC storage (Figure 1). Surface and deep SOC storages were significantly correlated with stand age (*R*^2^ = 0.758, *p* = 0.048; *R*^2^ = 0.817, *p* = 0.025; *n* = 7).

Note: 9 years, 10 years, 14 years, 26a years, 26b years, 28 years, and 30 years stand for the mixed plantation stands at the age of 9 years, 10 years, 14 years, 26 years, 26 years, 28 years, and 30 years, respectively. RWa, RWb, and RWc stand for reference wastelands in Tuobuka town, Pisha town, and Hulukou town, respectively. Data are means ± stand error, with the same letters on the same type of rectangles among stands denote insignificant differences among the same soil layer of stands (*α* = 0.05).

In comparison to adjacent reference wasteland, SOC accumulation rate in 0–80 cm soil profile in mixed plantations was calculated, ranging from 0.70 to 2.18 t C ha^−1^ yr^−1^, with an average value of 1.28 t C ha^−1^ yr^−1^. SOC accumulation rates at 0–20 cm soil depth ranged from 0.46 to 1.29 t C ha^−1^ yr^−1^, with an average of 0.75 t C ha^−1^ yr^−1^ (Figure 2). The accumulation rates of organic carbon at 20–80 cm depth ranged from 0.24 to 0.89 t C ha^−1^ yr^−1^, with an average of 0.53 t C ha^−1^ yr^−1^ (Figure 2). About 34–44% of the SOC accumulation to 80 cm depth was found at 20–80 cmdepth in mixed plantations. SOC accumulation rate deep into rock bed (0–60 cm soil profile) of reference natural recovery shrub grassland was 0.22 t C ha^−1^ yr^−1^, and 50% of SOC accumulation was found at 20–60 cm soil depth.

Note: 9 years, 10 years, 14 years, 26a years, 26b years, 28 years, and 30 years stand for the mixed plantation stands at the age of 9 years, 10 years, 14 years, 26 years, 26 years, 28 years, and 30 years, respectively. Percentages within the green and blue rectangles denote the proportion of SOC sequestration rate at 0–20 cm soil layer and at 20–80 cm soil layer, respectively, in 0–80 cm soil profile.

### 3.2. Biomass Carbon Storage

Total biomass carbon storage of 30-year-old mixed plantation was 77.78 t C ha^−1^, 54 times reference wasteland and 9 times reference natural recovery shrub-grassland (Figure 3). Biomass carbon sequestration rate of the 30-year-old mixed plantation was 2.54 t C ha^−1^ yr^−1^. Though tree biomass carbon stocks (including roots, stem, branches, and leaves) of the 30-year-old mixed plantation was significantly lower than that of reference natural forest, its biomass carbon stocks in shrub layer were significantly higher than that of reference natural forest for large quantities of shoots in the understory of the mixed plantation. Total biomass carbon storage in the mixed plantation was not significantly different from that in reference natural forest (79.26 t C ha^−1^) (*p* = 0.429) (Figure 3) and almost recovered to reference level after 30-year reforestation.

Though tree biomass carbon storage was the main component of total biomass carbon storage in both 30-year-old mixed plantation and reference natural forest, the secondary biomass carbon still accounts for 17% and 14%, respectively, in mixed plantation and reference natural forest. The rank of secondary biomass carbon stocks in the 30-year-old mixed plantation was as follows: litter compartment > shrub layer > coarse woody debris compartment > herb layer. The rank of secondary biomass carbon stocks in reference natural forest was as follows: litter compartment > coarse woody debris compartment > shrub layer > herb layer.

Note: Plantation: 30-year-old mixed plantation; RNF: reference natural forest; RW: reference wasteland in Hulukou town; RNS: reference natural recovery shrub-grassland. Data are means ± stand error, with the same letters among stands denoting insignificant differences (*α* = 0.05).

### 3.3. Total Ecosystem Carbon Sequestration

Total ecosystem carbon storage in the 30-year-old mixed plantation was 185.50 t C ha^−1^, 2.38 times reference wasteland, 2.29 times reference natural shrub grassland, and 29% lower than reference natural forest (Figure 4). Most ecosystem carbon was stored in the soil. Carbon pool contribution of soil, tree layer, and secondary biomass carbon was 58%, 35%, and 7%, respectively. In reference natural forest, total ecosystem carbon storage was 262.62 t C ha^−1^, and carbon pools of soil, tree layer, and secondary biomass accounted for 70%, 26%, and 4%, respectively (Figure 4). Total ecosystem carbon storage of reference natural recovery shrub grassland was 80.87 t C ha^−1^, and most of the carbon was in soil (90%) with a smaller amount in biomass (10%) (Figure 4). Total ecosystem carbon accumulation rate of the mixed plantation was 3.59 t C ha^−1^ yr^−1^. Most of the C accumulation was in biomass (71%) with a smaller amount in soil (29%). 

Note: Plantation: 30-year-old mixed plantation; RNF: reference natural forest; RW: reference wasteland in Hulukou town; RNS: reference natural recovery shrub-grassland. Data are means ± stand error, with the same letters among stands denoting insignificant differences (*α* =0.05).

## 4. Discussion

### 4.1. Total Soil Organic Carbon Accumulation in the Soil Profile After 30 Years of Reforestation

The average SOC accumulation rate in 0–80 cm soil profile in this research was 1.28 t C ha^−1^ yr^−1^ (Figure 2). Total SOC accumulation rate in our study was comparable to the effects of 34-year-old *Pinus taeda* and *Eucalyptus grandis* plantations on soil carbon in former miombo forest soils. After 34 years of plantation, net accumulation of SOC was 1.41 t C ha^−1^ yr^−1^ in *P. taeda* and 1.53 t C ha^−1^ yr^−1^ in *E. grandis* stands [21]. But the total SOC accumulation rate in this study was higher than that of the 24-year-old *P**. massoniana* plantation for top one-meter depth in the eroded red soil areas [15]. Surface SOC accumulation rate in this study ranged from 0.46 to 1.29 t C ha^−1^ yr^−1^ (Figure 2). It is also higher than other reports in the same region [16,22,23]. Tang and Li found that in the dry hot valley, surface SOC sequestration rate of 20-year-old *L. leuc**ocephala* plantation was 0.47 t C ha^−1^ yr^−1^ [22]. Tang et al. found that in 19-year-old *Acacia auriculiformis* plantation, SOC accumulation rates at the soil depth of 0–15 cm and 15–30 cm were 0.38 and 0.17 t C ha^−1^ yr^−1^, respectively [23]. In Yuanmou dry hot valley, SOC sequestration rate in 10-year-old *L. leuc**ocephala* plantation was only 0.05 t C ha^−1^ yr^−1^ [16]. Our results showed that reforestation with mixed species on former wasteland could enhance SOC sequestration. 

The relatively high rate of SOC accumulation during reforestation in this study was possibly due to good artificial restoration measures. Firstly, considering the niche distribution of different plant species in vertical and horizontal level, different trees and shrubs were mix-planted to avoid strong competition among species with similar niche. At the same time, the canopy of shrubs is close to ground, decreasing wind speed greatly, benefiting to litter accumulation, decreasing soil erosion, and increasing SOC input. Secondly, *L. leucocephala*, as the dominant species of mixed plantations is a fasting-growing woody species with massive, thin, and microphyllous litter. Production and properties of litter have significant effects on changes in SOC sequestration [24,25]. Further, in our study, *L. leucocephala* and initially planted species, *Cajanus cajan* and *Tephrosia candida,* were nitrogen-fixing plants. Some researchers have found that nitrogen fixation produced by nitrogen-fixing plants can reduce the decomposition of both old and new organic carbon through chemical stability and inhibiting microbial activity [26,27,28,29]. Lastly, mixed plantations in this study showed a reverse J-shaped size class of stem diameters, and more trees were in the smaller size classes. The recruitment in the multi-aged stand can supplement litterfall and fine root input in soil, maintaining high C sequestration rates for a very long period [29,30]. Therefore, niche complementary, good stand structure, and suitable plant species are important for SOC sequestration during reforestation.

Our results showed that SOC storage in mixed plantation after 30 years of reforestation was significantly lower than that of reference natural forest, which is different from the hypothesis. Some other reports reported that long-term reforestation restored SOC stock to the reference levels of the reference natural forest. Hu et al. (2018) found that in a subtropical karst region, SOC storage in 0–50 cm soil profile would recover to the primary forest level after 74 years [31]. It will take 600 years for Douglas-fir plantation to recover to the original soil carbon storage level [32]. Therefore, reference natural forest is an important SOC pool and should be protected strictly.

### 4.2. Deep Soil Organic Carbon Accumulation After 30 Years of Reforestation

This research suggested that deep soil in mixed plantations of the dry hot valley was important for SOC sequestration, with its contribution to total SOC sequestration ranging from 34% to 44% (Figure 2), and deep SOC storage increased with stand age significantly. This is possibly related with fine roots vertical distribution of dominant species *L. leucocephala*. Fine roots of *L. leucocephala* show a deep-seated growth trend with tree age. Fine root biomass at 20–100 cm depth of 3-, 5-, 9-, 14-, and 20-year-old *L**. leucocephala* contributes 58%, 53%, 71%, 75%, and 71% of that at 0–100 cm depth, respectively [33]. Distribution of plant roots can affect the vertical distribution of SOC [34], and fine root turnover and exudates of dominant plant species can significantly influence deep SOC storage [35,36]. Some researches indicate that in growing seasons, the amount of SOC input through fine root turnover may be equal to or even higher than that through leaf litter [37]. In addition to SOC input for fine root, vertical transportation of dissolved organic C and retention were important sources of deep SOC [38]. Besides, the contribution of deep soil to total SOC accumulation in the dry hot valley may be attributed to climates. Climates between dry and rain season vary greatly, and drying and wetting cycles increase soil organic carbon mineralization [39]. But deep soil is less influenced by drying and wetting cycles than surface soil and can facilitate deep SOC sequestration. Shi and Cui (2010) also found that a stable environment in the deep soil of the arid area was beneficial to SOC accumulation [40]. The investigation of deep SOC can effectively reveal the variation in soil carbon over several decades [41]. However, in the dry hot valley, deep SOC was usually overlooked in the past. Our results suggest that shallow topsoil sampling alone is insufficient for estimates of SOC accumulation after reforestation, and deep SOC accumulation should be taken into account for the assessment of total SOC change of the soil profile, especially in long-term studies. 

### 4.3. Total Biomass Carbon Sequestration

As hypothesized, our results demonstrated that biomass of mixed plantation was an important carbon stock and recovered to reference level of reference natural forest after 30 years of reforestation in dry hot valley of the Jinsha River. Biomass carbon storage in the 30-year-old mixed plantation was 77.78 t C ha^−1^ (Figure 3). It was higher than previous reports of younger plantations in this area. The biomass carbon stocks of *L. leucocephala* plantation and *Eucalyptus camaldulensis* – *L. leucocephala* plantation in Yuanmou dry hot valley were 20 t C ha^−1^ and 23 t C ha^−1^ at the stand age of 14 years [42] and 63.9 t C ha^−1^ and 41.5 t C ha^−1^ at the stand age of 20 years, respectively [43]. The high biomass carbon storage of mixed plantation in our study might be attributed to a long restoration period firstly. The mature mixed plantation in this study was 30 years old. The previous study indicated that biomass carbon storage in plantations is most strongly affected by stand age, followed by climate, biodiversity, and stand structural attributes [44]. Secondly, the dominant species of mixed plantations in this study was *L. leucocephala*, a pioneer species with characteristics of the fast-growing and high regeneration rate. Besides, the mature plantation in this study was mixed species with niche complementary and good community structure. Li et al. (2019) indicated that niche complementary, the ability of hyperdiverse communities to better use the limited resources, might be the mechanism of promoting tree carbon storage in subtropical forests [44]. DBH of the mixed plantation showed a reversed J-shaped size class pattern in our research. More trees were in the smaller size classes, and the mixed plantation in our study had high primary productivity. This study showed that niche complementary, good stand structure, and dominant species *L. leucocephala* facilitated total biomass carbon sequestration of the mixed plantations in dry hot valley. 

In our study, secondary biomass carbon accounted for 17% and 14% of total biomass carbon storage in mature mixed plantation and reference natural forest, respectively. The result was comparable to some studies. Zhang et al. (2012) found that secondary biomass carbon in *Eucalyptus* and *Acacia* plantations accounted for 10.2% and 20.3% of total biomass carbon, respectively, in the subtropical area [13]. Secondary biomass carbon was 8.2–18.4% of total biomass carbon in subalpine plantations in southwestern China [45]. This study demonstrated that secondary biomass carbon could not be overlooked.

### 4.4. Total Ecosystem Carbon Pool 

Total ecosystem carbon stock of 30-year-old mixed plantation was 185.50 t C ha^−1^, 29% lower than that of reference natural forest, and the biggest component was SOC, accounting for 58% (Figure 4). This result was comparable to the result of Veloso et al. (2018). Veloso et al. (2018) found that soil held 48–59% of the total ecosystem carbon in pine plantations [3]. In this study, the total ecosystem carbon sequestration rate of the 30-year-old plantation was 3.59 t C ha^−1^ yr^−1^. Our result was lower than the report in the subtropical area [12]. Lewis et al. (2019) found that carbon sequestration rate in Australian subtropical area was 7.4 t C ha^−1^ yr^−1^ and predicted that it would take nearly 50 years for C stocks to reach a similar level of reference natural ecosystems [12]. This study suggested that it was prospective to recover total ecosystem carbon to reference level by reforestation with *L. leucocephala* and other species.

This study showed that most of the C accumulation of 30-year-old mixed plantation was in biomass (71%), with a smaller amount in soil (29%), which is different from the hypothesis. It is similar to the result of Lewis et al. (2019). Lewis et al. (2019) indicated that C sequestration associated with reforestation was mainly through above- and below-ground biomass sequestration in the subtropical area after reforestation on croplands, and biomass carbon sequestration accounted for 82% [12]. Our results showed that biomass carbon sequestration during reforestation was more rapid than SOC sequestration in the dry hot valley.

In comparison to mixed plantations and reference natural recovery shrub grassland, reference natural forest was the most important ecosystem carbon pool. Total ecosystem carbon storage was 262.61 t C ha^−1^, which was 184.75 t C ha^−1^, 181.73 t C ha^−1^, and 77.11 t C ha^−1^ higher than that of adjacent reference wasteland, reference natural recovery shrub grassland, and 30-year-old mixed plantation, respectively (Figure 4). The reasons for high total ecosystem carbon storage in reference natural forest were as follows: The reference natural forest was protected as “Fengshui” forest, disturbed lightly for some collections. Further, reference natural forest had good structure, with well-developed understory and a large quantity of litter in the understory, which could increase rainwater infiltration into the soil in rain seasons, reducing water evaporation in dry seasons. Dense litterfall could make soil maintain good soil moisture, reduce the time below the plant withering moisture, and facilitate plant growth and SOC input. Therefore, the natural forest should be protected. Once the natural forest was destroyed and degraded into wastelands, a large quantity of carbon would be released into the environment.

## 5. Conclusions

In the dry hot valley, reforestation with mixed species on former wastelands can significantly enhance surface and deep SOC sequestration. Deep SOC storage of mixed plantations increased with stand age. Deep soil contribution to the total SOC sequestration was up to 34–44%. It is possibly due to an increase in root exudates and vertical transportation of dissolved organic C and retention into deep soil during reforestation. After 30 years of reforestation, biomass carbon of mixed plantations recovered to reference level, and total ecosystem carbon storage was 2.38 times reference wasteland, 2.29 times reference natural shrub grassland, and 29% lower than reference natural forest. In 30-year-old mixed plantation, SOC was the biggest component of total ecosystem carbon, but biomass carbon sequestration contribution accounted for most. It indicated that active reforestation could facilitate more rapid carbon sequestration, especially biomass carbon than reference natural recovery in the dry hot valley. Niche complementary, good stand structure, and dominant species *L. leucocephala* played an important role in the faster rate of C sequestration in dry hot valley of the Jinsha River.

## Figures and Tables

**Figure 1 ijerph-16-01937-f001:**
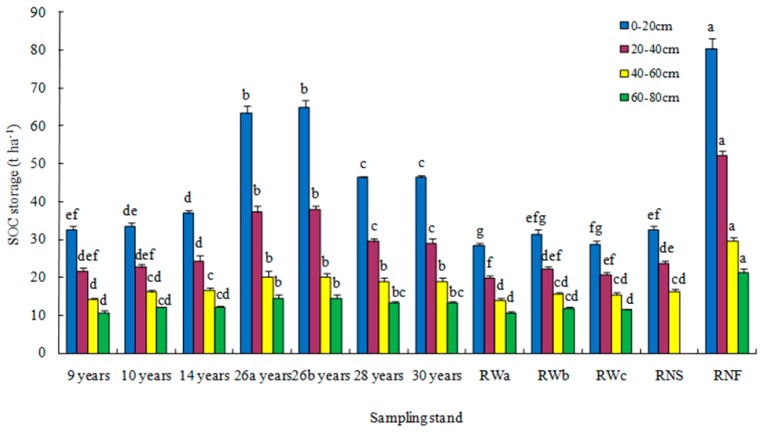
Soil organic carbon (SOC) storage at different depths in the soil profile of mixed plantations, reference wastelands, reference natural recovery shrub grassland (RNS), and reference natural forest (RNF) (t C ha^−1^).

**Figure 2 ijerph-16-01937-f002:**
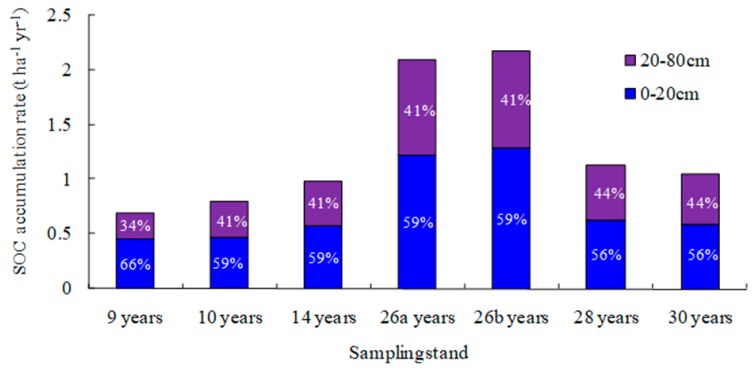
Soil organic carbon (SOC) accumulation rate of mixed plantations, including surface soil (0–20 cm) and deep soil (20–80 cm).

**Figure 3 ijerph-16-01937-f003:**
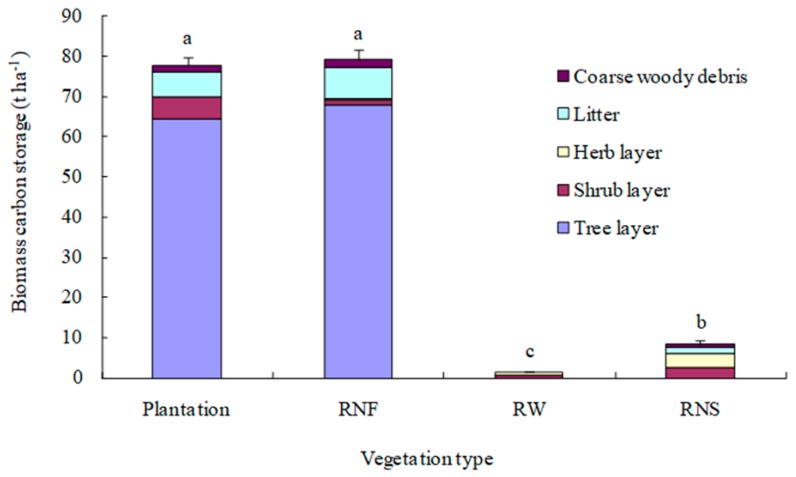
Biomass carbon storage of different vegetation types.

**Figure 4 ijerph-16-01937-f004:**
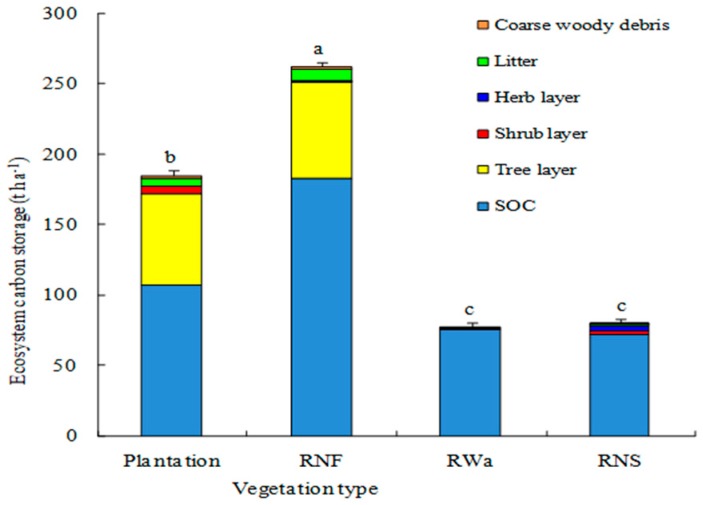
Ecosystem carbon storage of different vegetation types, including soil organic carbon (SOC), tree biomass carbon, and secondary biomass carbon.

**Table 1 ijerph-16-01937-t001:** Basic information of the stands.

Sites	Stands	Slope aspect (°)	Slope (°)	Altitude (m)	Woody species richness	Woody species heterogeneity
Tuobuka town, Dongchuan municipality, Yunnan (N26°25′12″; E103°04′43″)(MAT: 22 ℃; MAP: 700 mm)	9 years old mixed plantation (9 years)	NE80	20	895	S_T_ = 3; S_S_ = 4	*H’_T_* = 0.15; *H’_S_* = 0.27
	Reference wasteland in Tuobuka (RWa) (being wasteland for over 30 years)	NE80	20	910	S_S_ = 1	*H’_S_* = 0
Pisha town, Ningnan county, Sichuan (N27°04′15″; E102.43′42″) (MAT: 20 ℃; MAP: 800 mm)	26 years old mixed plantation (26a years)	NE75	20	1273	S_T_ = 2; S_S_ = 3	*H’_T_* = 0.17; *H’_S_* = 0.23
	26 years old mixed plantation (26b years)	NE76	19	1273	S_T_ = 1; S_S_ = 3	*H’_T_* = 0; *H’_S_* = 0.20
	Reference wasteland in Pisha (RWb) (being wasteland for over 46 years)	NE77	22	1260	S_S_ = 1	*H’_S_* = 0
	Reference natural forest (NF) (about 200 years old)	NE45	27	1230	S_T_ = 9; S_S_ = 15	*H’_T_*= 1.57; *H’_S_* = 2.28
Hulukou town, Ningnan county, Sichuan (N26°57′24″; E102.53′01″) (MAT: 22 ℃; MAP: 700 mm)	10 years old mixed plantation (10 years)	NE63	18	860	-	-
	14 years old mixed plantation (14 years)	NE66	19	800	-	-
	28 years old mixed plantation (28 years)	NE60	17	840	S_T_ = 1; S_S_ = 3	*H’_T_* = 0; *H*’*_S_* = 0.36
	30 years old mixed plantation (30 years)	NE70	18	805	S_T_ = 3; S_S_ = 5	*H’_T_* = 0.26; *H’_S_* = 0.44
	Reference natural recovery shrub grassland (RNS) (about 35 years old)	NE63	21	840	S_S_ = 4	*H’_S_* = 0.46
	Reference wasteland in Hulukou (RWc) (being wasteland for over 55 years)	NE35	21	821	S_S_ = 1	*H’*_S_ = 0

Note: MAT: mean annual temperature; MAP: mean annual precipitation; S_T_: species richness in tree layer; S_s_: species richness in shrub layer; *H’_T_*: Shannon-Winner index in tree layer; *H’_S_*: Shannon-Winner index in the shrub layer.

**Table 2 ijerph-16-01937-t002:** Soil parameters, such as pH and soil bulk density, at each stand.

Sites	Stands	pH				Bulk density (g/cm^3^)			
		0–20 cm	20–40 cm	40–60 cm	60–80 cm	0–20 cm	20–40 cm	40–60 cm	60–80 cm
Tuobuka town	9 years	7.6	7.5	7.9	8.0	1.52	1.57	1.59	1.61
	RWa	7.9	7.8	8.0	8.0	1.57	1.59	1.60	1.61
Pisha town	26a years	5.7	5.6	5.8	5.9	1.36	1.41	1.43	1.45
	26b years	6.8	6.7	7.0	7.0	1.33	1.39	1.42	1.44
	RWb	7.5	7.4	7.7	7.7	1.43	1.45	1.46	1.47
	RNF	5.6	5.5	5.6	5.8	1.35	1.37	1.41	1.41
Hulukou town	10 years	7.6	7.5	7.7	8.0	1.53	1.57	1.59	1.59
	14 years	8.0	7.9	8.0	8.0	1.47	1.55	1.58	1.59
	28 years	7.4	7.3	7.3	7.6	1.44	1.51	1.55	1.58
	30 years	7.7	7.6	7.7	7.9	1.44	1.52	1.56	1.59
	RNS	7.8	7.7	8.0	-	1.47	1.54	1.57	-
	RWc	8.2	8.1	8.1	8.2	1.57	1.59	1.60	1.61

Note: 9 years, 10 years, 14 years, 26a years, 26b years, 28 years, and 30 years stand for the mixed plantation stands at the age of 9 years, 10 years, 14 years, 26 years, 26 years, 28 years, and 30 years, respectively. RWa, RWb, and RWc stand for reference wastelands in Tuobuka town, Pisha town, and Hulukou town, respectively. RNF and RNS stand for reference natural forest and reference natural recovery shrub grassland, respectively.

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
