# Peer review of "Rapid Sequestration of Ecosystem Carbon in 30-year Reforestation with Mixed Species in Dry Hot Valley of the Jinsha River"

_ijerph, 2019, doi:10.3390/ijerph16111937_

Round 1

Reviewer 1 Report

The topic and the result are quite interesting. However, the authors need to present better their results. See comments in the attached file. In particular, the knowledge limit must be better explained in the abstract and the Introduction, because actually it looks like 'there has not been such studies in this valley', which is very local. You need to identify clearly what your results show versus actual literature results. Also, the abstract is much too long with too many results given. It should be focussed on one discovery/result of world interest. For instance how do the results below compare with actual literature?:

'Total biomass carbon storage in 30-year-old plantation was 77.78 t C ha-1, 54 times that of degraded wasteland, 9 times that of natural recovery shrub grassland and insignificantly lower than that of natural forest (p0.05). After 30 years reforestation, plantation biomass carbon storage recovered to original level and its sequestration rate was 2.54 t C ha-1 yr-1.'

Author Response

1.    General comments:

However, the authors need to present better their results. See comments in the attached file. In particular, the knowledge limit must be better explained in the abstract and the Introduction, because actually it looks like 'there has not been such studies in this valley', which is very local. You need to identify clearly what your results show versus actual literature results. Also, the abstract is much too long with too many results given. It should be focused on one discovery/result of world interest. For instance how do the results below compare with actual literature?

Reply: Accept and revision is done.

2.    Specific comments:

1)    This title is ok but a title focusing on the main result, advance or discovery would be better.

Reply: Accept and revision is done.

Title: “Rapid sequestration of ecosystem carbon in 30-year reforestation with mixed species in dry hot valley of the Jinsha River”

2)    This abstract is much too long with too many results given. A good abstract has about 3 sentences of global issues, scientific issues, knowledge limit/hypothesis; 3-4 sentences of experimental (samples, location, methods, parameters measured); and 3-4 sentences giving 1-3 results (no more) with data, the interpretation of these results, then the novelty/difference of those results versus actual knowledge. Report in the abstract ONLY the results that show something new versus actual literature.

Reply: Accept and revision is done.

3)    This issue/problem is true, but 'local' (no such study in this valley). If you wish to interest world readers you need to identify a knowledge limit of world

Reply: Accept and revision is done.

4)    the knowledge limit must be better explained in the abstract and the Introduction, because actually it looks like 'there has not been such studies in this valley', which is very local.

Reply: Accept and revision is done.

5)    “deep soil played an important role” is too vague.

Reply: Accept and revision is done.

6)    This looks as a major result that I would keep in the abstract.

Reply: Accept.

7)    no abbreviation in titles

Reply: Accept and revision is done.

8)    Are these your results? In all the text of the manuscript, you need to identify very clearly when you write about your results, by using the first person (We found, Our results show...) or other expresssions (In this study, Figure 1, Table 1...)

Reply: Accept and revision is done.

9)    “The averaged SOC accumulation rate in 0-80 cm soil profile in this research was 1.28 t C ha-1 yr-1”in which figure/table?

Reply: in Fig.2. And the revision is done.

10)  too long paragraph. A paragraph must show only one story of demontration.

Reply: Accept and revision is done.

11)   This discussion is too long. This is not a 'long review' of the literature, this is the 'discussion of your results'. You just need to explain what are the difference between your results (figure X) and the literature.

Reply: Accept and revision is done.

12)  'in this study' and placing a reference '17' is contradictory: if these are your results, they should not be in the literature.

Reply: Accept and revision is done.

13)  All subsections should end by a firm conclusion on Your results: insterpretation, meaning, novelty.

Reply: Accept and revision is done.

14)  here you are mixing your results with literature results in the same sentence: never do that because the reader will be confused. Give briefly your result in one sentence (fig X). Then discuss this result by comparision with the literature. Apply everywhere

Reply: Accept and revision is done.

15)  never end a subsection by references. see previous remark

Reply: Accept and revision is done.

Reviewer 2 Report

This is an interesting study that provides useful information regarding the effects on carbon sequestration of reforestation in the dry-hot valley of the Jinsha River. Overall, the manuscript is clear and concise, and the findings provide insights into the benefits and recovery time expectations for carbon stocks following reforestation in the region and areas under similar environmental constraints. The insights into deep soil carbon accumulation are particularly interesting because the deep soil is often ignored. My primary concern regarding the study is the ability of the reference natural recovery grassland and the reference natural forest to be used as references for all the plantations when they are not adjacent to most of the plantations. My comments are below.

GENERAL COMMENTS

Given that the reference natural recovery grassland and the reference natural forest are not adjacent to most of the plantations, these results should be used with great caution, unless the authors can show that the soil types, climate variables, and length of natural recovery were similar and comparable. For example, Figure 2 suggests that the site where the 26-year-old stands and the reference natural forest were was potentially not comparable to the other sites, as one would not expect such a drastic gain in SOC accumulation over 12 years, and especially not such a drastic decrease in just two years. The comparisons between the adjacent wastelands and the plantations are thus more informative and should be emphasized.

Use consistent and clear terminology throughout the manuscript. For example, reforestation and afforestation are both used; if they are being used interchangeably, choose one term, but if they are being used to describe different circumstances, clearly define the differences. The same is true for “mixed plantations” and “plantation forests;” choose the most appropriate and descriptive term and use it consistently, or it is confusing to the reader. Additionally, certain terms used are not well-defined, such as “secondary biomass.” Make sure to define terms the first time they are used, such as defining “surface soil” as 0-20 cm and “deep soil” as 20-80 cm for the context of this study.

Make sure to spell out an abbreviation the first time it’s used, and vice versa, and to be consistent from that point forward with the use of the abbreviation. Try to be consistent with sentence structure, especially when providing lists or similar/related information. For example, when describing differences in values, do not use both multiplier differences and percentage differences in the same sentence.

Provide actual p-values, as opposed to, for example (p<0.05).

Moderate revisions for English and edits for grammatical errors are needed.

Please use numbered sentences for all future manuscript submissions.

SPECIFIC COMMENTS

Title: I suggest revising the title to reflect the key take away point(s) of the study.

Abstract:

-Second and third sentence need to be more clear, precise, and informative about why the study is needed and the context of the study. There’s no need to list what will be measured because it is overviewed later

-The increase in deep SOC with stand age is interesting and worth noting with a brief mention of the possible contribution of fine root exudates and DOC to this SOC stock

-In general, the methods and results need to be clearer and more concise; also see comments below regarding methods, results, and conclusions

Introduction:

-Near end of first paragraph: “…deep SOC may be more stable…” Please revise the use of the term “stable.” The factors mentioned facilitate longer-term storage, but do not make the SOC intrinsically stable. See: Gross, C. D.; Harrison, R. B. The case for digging deeper: Soil organic carbon storage, dynamics, and controls in our changing world. Soil Syst. 2019, 3(2), 28, doi:10.3390/soilsystems3020028 (also relevant for discussion at top of page 10)

-Near beginning of third paragraph: “…clearly demonstrated…” Can you briefly describe how it was clearly demonstrated (i.e., some results of the study)?

-Near end of first paragraph on page 2: “…natural forests are badly destroyed.” This is unclear. Please better describe the state of the forests in the region, as well as the wastelands, naturally recovering grasslands, and plantations. Some brief history would be useful. Also, please provide the goal of the plantations (i.e., to replicate natural forest or for future harvest?).

-Last paragraph of intro: Please make this paragraph clearer and more concise

Materials and Methods:

-Were plots selected randomly?

-For clarity, better to consistently use “reference” in front of both the natural recovery grassland and natural forest stand; also, please be clearer in the text about the fact that only the reference wastelands are adjacent to each plantation, whereas the reference natural recovery grassland and natural forest are single sites against which all plantations were compared

-How might the lack of a reference natural recovery grassland and natural forest for each plantation bias the results? Is it fair to compare all the plantations to these reference sites? Why or why not?

-Please include MAP and MAT climate variables in Table 1, as well as length of time each wasteland was a wasteland, the age of the natural forest, and the length of the natural recovery of the grassland; also, the table is a bit inconsistent with the information provided for each stand (e.g., 28-year-old stand) and “reference” is missing for the natural recovery grassland; provide list of abbreviations for the table

-Please include a table of general soil properties at each site/stand (texture, pH, bulk density, etc.)

-Last sentence of first paragraph: Is this the distance between plots within a stand? What was the max distance between plots? How distant were reference stands from plantations (max and min)? Were the adjacent wastelands for each plantation part of the same wasteland on which the plantation was grown? Was the reference natural recovery grassland part of the wasteland for that site as well?

-The introduction, as well as most of the relevant studies, emphasize studying deep soil to 100 cm; why was the soil only sampled to 80 cm? Also, how many pits hit bedrock before 80 cm?

-What method was used to take the soil samples?

-Please briefly describe the bucket method for bulk density

-Regarding using bulk density for SOC stock calculations, caution should be used because it is likely bulk density varies between sites and changed with vegetation. Therefore, when sampling by depth, more or less soil may be sampled based on the direction of change in bulk density. How substantially did bulk density differ between vegetation types and did this change affect the overall SOC stock results? See: Wendt, J.W., and S. Hauser. 2013. An equivalent soil mass procedure for monitoring soil organic carbon in multiple soil layers. Eur. J. Soil Sci. 64:58–65. doi:10.1111/ejss.12002

-It would be interesting and useful to also calculate the SOC accumulation rate for the naturally recovery grasslands

-Were the 6 trees that represented the mean size collected at random from all the plots? Also, it is unclear how the measurements from the destructive tree sampling were used. Please clarify this. The previous sentence suggests that tree heights and DBH were used to estimate tree layer biomass C. Were all trees in the plots measured for height and DBH?

-Briefly define coarse woody debris and how it was distinguished from the litter layer

-Were shrub and grass root C biomass accounted for? It seems that, without accounting for this pool, a substantial underestimation in biomass C at those sites is likely

-Were the assumptions for ANOVA met?

Results:

-No need to separate subsection 3.1 into sub-subsections

-Please be consistent with layers; otherwise, it seems that the data was cherry-picked to find groups of layers that were statistically different from the other groups; the layers should be analyzed by the 20 cm increments in which they were sampled; additional logical analyses include total (0-80 cm) and surface (0-20 cm) and deep (20-80 cm) soil as defined by the study

-Fig. 1: RWa; standard deviation (but why not standard error?); the letter designation is confusing – I believe the same depth layers across stands are being compared; provide consistent information for all stands in the figure caption, and provide list of abbreviations; α = 0.05

-Reiterate that the accumulation rates are in comparison to the adjacent wasteland

-Fig. 2: same comments as Fig. 1; remove statement about letters and significance; add actual percentages within green and blue rectangles and make a statement about this in the figure caption

-First sentence of 2nd paragraph of 3.2: Suggests tree biomass C stock was greater than SOC stock

-Clarify whether “tree biomass” includes roots

-Biomass C stocks returning to “original” level is misleading; should be “reference” level

-Last sentence of 2nd paragraph of 3.2: This is a conclusion

-Figures 3 and 4: Same comments as Figures 1 and 2, where applicable

Discussion:

-First paragraph is too long; break it up logically

-First paragraph: Report SOC accumulation rates of references 22-24 for reference

-Sentence in first paragraph: “Besides, litterfall of…soil nutrient conditions.” Citation?

-Subsection 4.3 first sentence: Biomass C stocks returning to “initial” level is misleading; should be “reference” level

-Top of page 11: Can the authors provide an estimate of recovery time of total ecosystem C? “A long time” is not a defined or useful conclusion

-Final paragraph that starts with “In comparison…” In comparison to what?

Conclusions:

-Biomass C stocks returning to “original” level is misleading; should be “reference” level

-The increase in deep SOC with stand age is interesting and worth noting with a brief mention of the possible contribution of fine root exudates and DOC to this SOC stock

-Final sentence: Since this study only looked at plantations with mixed species including L. leucocephala, the conclusion cannot be drawn that this is better or worse than other plantation stand structures and tree species; the authors could suggest that this stand structure and L. leucocephala played a role in the faster rate of C sequestration, as argued in the discussion

References:

-Remove double set of numbers

Author Response

Dear Editors and Reviewers:

Revised portion are marked using the "Highlight" function in the paper. The main corrections in the paper and the responds to the reviewer’s comments are as flowing:

Responds to the reviewer’s comments:

1 GENERAL COMMENTS

1)    Given that the reference natural recovery grassland and the reference natural forest are not adjacent to most of the plantations, these results should be used with great caution, unless the authors can show that the soil types, climate variables, and length of natural recovery were similar and comparable. For example, Figure 2 suggests that the site where the 26-year-old stands and the reference natural forest were was potentially not comparable to the other sites, as one would not expect such a drastic gain in SOC accumulation over 12 years, and especially not such a drastic decrease in just two years. The comparisons between the adjacent wastelands and the plantations are thus more informative and should be emphasized.

Reply:

----The information of reference natural recovery shrub grassland and reference natural forest was added in the thesis: “For limited area of natural forests and natural recovery shrub grasslands with similar topography and edaphic conditions to those of mixed plantations in dry hot valley, only one reference natural recovery shrub grassland (located in Hulukou town, and about 1.0-1.5 km away from adjacent plantations ) and one reference natural forest (located in Pisha town, and about 2.5 km away from adjacent plantations) were selected. Although to some extent there is limitation for lack of a reference natural recovery grassland and a reference natural forest for each plantation, it is feasible to compare all the plantations to these reference sites because all the stands are in dry hot valley, with similar temperature, precipitation and soil types. The soil is dry red soil [Chromic Cambisol in Food and Agriculture Organization (FAO) taxonomy, Ustochrept in United States Department of Agriculture (USDA) taxonomy]. Mean annual temperature is 20-22 and average annual rainfall is 700-800 mm.”

----It is also emphasized that SOC accumulation rate in each plantation was calculated as the difference in SOC storage between each plantation and the reference wasteland, divided by the recovery time.

2)    Use consistent and clear terminology throughout the manuscript. For example, reforestation and afforestation are both used; if they are being used interchangeably, choose one term, but if they are being used to describe different circumstances, clearly define the differences. The same is true for “mixed plantations” and “plantation forests;” choose the most appropriate and descriptive term and use it consistently, or it is confusing to the reader. Additionally, certain terms used are not well-defined, such as “secondary biomass.” Make sure to define terms the first time they are used, such as defining “surface soil” as 0-20 cm and “deep soil” as 20-80 cm for the context of this study.

Reply: Accept and revision is done.

3)    Make sure to spell out an abbreviation the first time it’s used, and vice versa, and to be consistent from that point forward with the use of the abbreviation. Try to be consistent with sentence structure, especially when providing lists or similar/related information. For example, when describing differences in values, do not use both multiplier differences and percentage differences in the same sentence.

Reply: Accept and revision is done.

4)    Provide actual p-values, as opposed to, for example (p<0.05).

Reply: Accept and revision is done.

5)    Moderate revisions for English and edits for grammatical errors are needed.

Reply: Accept and revision is done.

6)    Please use numbered sentences for all future manuscript submissions.

Reply: Accept and revision is done.

2 SPECIFIC COMMENTS

1)    Title: I suggest revising the title to reflect the key take away point(s) of the study.

Reply: Accept and revision is done.

Title: Rapid sequestration of ecosystem carbon in 30-year reforestation with mixed species in dry-hot valley of the Jinsha River.

2)    Abstract:

(1) Second and third sentence need to be more clear, precise, and informative about why the study is needed and the context of the study. There’s no need to list what will be measured because it is overviewed later

Reply: Accept and revision is done.

(2) The increase in deep SOC with stand age is interesting and worth noting with a brief mention of the possible contribution of fine root exudates and DOC to this SOC stock.

Reply: Accept and revision is done.

(3) In general, the methods and results need to be clearer and more concise; also see comments below regarding methods, results, and conclusions

Reply: Accept and revision is done.

3)    Introduction:

(1) Near end of first paragraph: “…deep SOC may be more stable…” Please revise the use of the term “stable.” The factors mentioned facilitate longer-term storage, but do not make the SOC intrinsically stable.

Reply: Accept and revision is done.

(2) Near beginning of third paragraph: “…clearly demonstrated…” Can you briefly describe how it was clearly demonstrated (i.e., some results of the study)?

Reply: Accept and revision is done

(3) Near end of first paragraph on page 2: “…natural forests are badly destroyed.” This is unclear. Please better describe the state of the forests in the region, as well as the wastelands, naturally recovering grasslands, and plantations. Some brief history would be useful. Also, please provide the goal of the plantations (i.e., to replicate natural forest or for future harvest?).

Reply: Accept and revision is done.

(4) Last paragraph of intro: Please make this paragraph clearer and more concise.

Reply: Accept and revision is done.

4) Materials and Methods:

(1) Were plots selected randomly?

Reply: Plots were selected randomly.

(2) For clarity, better to consistently use “reference” in front of both the natural recovery grassland and natural forest stand

Reply: Accept and revision is done.

(3) please be clearer in the text about the fact that only the reference wastelands are adjacent to each plantation, whereas the reference natural recovery grassland and natural forest are single sites against which all plantations were compared

Reply: Accept and revision is done.

(4) How might the lack of a reference natural recovery grassland and natural forest for each plantation bias the results? Is it fair to compare all the plantations to these reference sites? Why or why not?

Reply: Although to some extent there is limitation for lack of a reference natural recovery grassland and natural forest for each plantation, it is feasible to compare all the plantations to these reference sites because all the stands are in dry hot valley, with similar temperature, precipitation and soil types.

The related information was supplemented in the text:

“For limited area of natural forests and natural recovery shrub grasslands with similar topography and edaphic conditions to those of mixed plantations in dry hot valley, only one reference natural recovery shrub grassland (located in Hulukou town, and about 1.0-1.5 km away from adjacent plantations) and one reference natural forest (located in Pisha town, and about 2.5 km away from adjacent plantations) were selected. Although to some extent there is limitation for lack of a reference natural recovery grassland and a reference natural forest for each plantation, they were located in the same area (dry hot valley) as all the plantations in this study. The soil is dry red soil [Chromic Cambisol in Food and Agriculture Organization (FAO) taxonomy, Ustochrept in United States Department of Agriculture (USDA) taxonomy]. Mean annual temperature is 20-22 and average annual rainfall is 700-800 mm.”

(5) Please include MAP and MAT climate variables in Table 1, as well as length of time each wasteland was a wasteland, the age of the natural forest, and the length of the natural recovery of the grassland; also, the table is a bit inconsistent with the information provided for each stand (e.g., 28-year-old stand) and “reference” is missing for the natural recovery grassland; provide list of abbreviations for the table

Reply: Accept and revision is done.

(6) Please include a table of general soil properties at each site/stand (texture, pH, bulk density, etc.)

Reply: Accept and revision is done.

(7) Last sentence of first paragraph: Is this the distance between plots within a stand? What was the max distance between plots? How distant were reference stands from plantations (max and min)? Were the adjacent wastelands for each plantation part of the same wasteland on which the plantation was grown? Was the reference natural recovery grassland part of the wasteland for that site as well?

Reply: Last sentence of first paragraph: This is the distance between plots within a stand. The information of distance between plots and between reference stands and plantations was supplemented in the text. The land of each mixed plantation prior to reforestation was the part of corresponding adjacent reference wasteland. and it is the same to reference natural recovery shrub grassland.

(8) The introduction, as well as most of the relevant studies, emphasize studying deep soil to 100 cm; why was the soil only sampled to 80 cm? Also, how many pits hit bedrock before 80 cm?

Reply: Because soil was about 90 cm deep in all stands except reference natural recovery shrub grassland (with soil depth of about 60 cm), three soil pits were excavated to 80 cm deep or to bedrock (in reference natural recovery shrub grassland) randomly in each plot.

(9) What method was used to take the soil samples? Please briefly describe the bucket method for bulk density

Reply: Accept and the method of taking soil sampes and bulk density method were supplemented in the text.

(10)Regarding using bulk density for SOC stock calculations, caution should be used because it is likely bulk density varies between sites and changed with vegetation. Therefore, when sampling by depth, more or less soil may be sampled based on the direction of change in bulk density. How substantially did bulk density differ between vegetation types and did this change affect the overall SOC stock results?

Reply: Although bulk density varies between sites and changed with vegetation, soil depth of all sampling stands was about 90 cm except natural recovery shrub grassland (60 cm depth). In our study, soil was sampled to 80 cm or to rockbed (in reference natural recovery shrub grassland), with little systematic influence caused by bulk density difference.

(11)It would be interesting and useful to also calculate the SOC accumulation rate for the naturally recovery grasslands

Reply: SOC accumulation rate of the naturally recovery grasslands was calculated in the result part.

(12)Were the 6 trees that represented the mean size collected at random from all the plots? Also, it is unclear how the measurements from the destructive tree sampling were used. Please clarify this. The previous sentence suggests that tree heights and DBH were used to estimate tree layer biomass C. Were all trees in the plots measured for height and DBH?

Reply:

---The 6 trees that represented the mean size were collected at random from all the plots within the stand.

--- In order to determine the height and diameter at breast height of the mean tree, all trees in the plots measured for height and DBH.

---The clearer information of destructive tree sampling was supplemented in the text.

(13)Briefly define coarse woody debris and how it was distinguished from the litter layer.

Reply: Accept and revision is done.

(14)Were shrub and grass root C biomass accounted for? It seems that, without accounting for this pool, a substantial underestimation in biomass C at those sites is likely.

Reply: Shrub and grass root were collected to 30 cm soil depth and account for biomass carbon.

(15)Were the assumptions for ANOVA met?

Reply: The assumptions for ANOVA were met and the information was supplemented in the text.

5) Results:

(1)No need to separate subsection 3.1 into sub-subsections

Reply: Accept and revision is done.

(2)Please be consistent with layers; otherwise, it seems that the data was cherry-picked to find groups of layers that were statistically different from the other groups; the layers should be analyzed by the 20 cm increments in which they were sampled; additional logical analyses include total (0-80 cm) and surface (0-20 cm) and deep (20-80 cm) soil as defined by the study

Reply: Accept and revision is done.

(3)Fig. 1: RWa; standard deviation (but why not standard error?); the letter designation is confusing – I believe the same depth layers across stands are being compared; provide consistent information for all stands in the figure caption, and provide list of abbreviations; α = 0.05

Reply: Accept and revision is done.

(4)Reiterate that the accumulation rates are in comparison to the adjacent wasteland.

Reply: Accept and revision is done.

(5)Fig. 2: same comments as Fig. 1; remove statement about letters and significance; add actual percentages within green and blue rectangles and make a statement about this in the figure caption

Reply: Accept and revision is done.

(6)First sentence of 2nd paragraph of 3.2: Suggests tree biomass C stock was greater than SOC stock

Reply: There is something wrong with this sentence and it is corrected: “Though tree biomass carbon storage was the main component of total biomass carbon storage in both 30-year-old plantation and reference natural forest”

(7)Clarify whether “tree biomass” includes roots

Reply: Accept and revision is done.

(8)Biomass C stocks returning to “original” level is misleading; should be “reference” level

Reply: Accept and revision is done.

(9)Last sentence of 2nd paragraph of 3.2: This is a conclusion

Reply: Accept and revision is done. The last sentence of 2nd paragraph of 3.2 is deleted.

(10)Figures 3 and 4: Same comments as Figures 1 and 2, where applicable

Reply: Accept and some revisions are done. Because some rectangles in fig.3 and 4 are so small that percentages can’t be added. Therefore, percentages are not added in fig. 3 and 4.

6) Discussion:

(1)First paragraph is too long; break it up logically

Reply: Accept and revision is done.

(2)First paragraph: Report SOC accumulation rates of references 22-24 for reference

Reply: Accept and revision is done.

(3)Sentence in first paragraph: “Besides, litterfall of…soil nutrient conditions.” Citation?

Reply: We mixed ours results and other literature result. The revision is done.

(4)Subsection 4.3 first sentence: Biomass C stocks returning to “initial” level is misleading; should be “reference” level

Reply: Accept and revision is done.

(5)Top of page 11: Can the authors provide an estimate of recovery time of total ecosystem C? “A long time” is not a defined or useful conclusion

Reply: Because total ecosystem carbon was measured only for one of those mixed plantations, recovery time of total ecosystem carbon can’t be estimated in this study. And the sentence was changed: “This study suggested that it was prospective to recover total ecosystem carbon to reference level by reforestation with L. leucocephala and other species.”

(6)Final paragraph that starts with “In comparison…” In comparison to what?

Reply: This sentence is revised: “In comparison to mixed plantations and reference natural recovery shrub grassland, reference natural forest was the most important ecosystem carbon pool.”

7) Conclusions:

(1)Biomass C stocks returning to “original” level is misleading; should be “reference” level

Reply: Accept and revision is done.

(2)The increase in deep SOC with stand age is interesting and worth noting with a brief mention of the possible contribution of fine root exudates and DOC to this SOC stock

Reply: Accept and revision is done.

(3)Final sentence: Since this study only looked at plantations with mixed species including L. leucocephala, the conclusion cannot be drawn that this is better or worse than other plantation stand structures and tree species; the authors could suggest that this stand structure and L. leucocephala played a role in the faster rate of C sequestration, as argued in the discussion

Reply: Accept and revision is done.

8) References:

(1)Remove double set of numbers

Reply: Accept and revision is done.

Special thanks to you for your good comments.